# In Situ Formation of Acidic Comonomer during Thermal Treatment of Copolymers of Acrylonitrile and Its Influence on the Cyclization Reaction

**DOI:** 10.3390/polym16192833

**Published:** 2024-10-07

**Authors:** Roman V. Toms, Daniil A. Ismaylov, Alexander Yu. Gervald, Nickolay I. Prokopov, Anna V. Plutalova, Elena V. Chernikova

**Affiliations:** 1Institute of Fine Chemical Technologies, MIREA—Russian Technological University, pr. Vernadskogo, 86, 119571 Moscow, Russia; star.bro900@gmail.com (D.A.I.); gervald@bk.ru (A.Y.G.); prokopov@mirea.ru (N.I.P.); 2Faculty of Chemistry, Lomonosov Moscow State University, Lenin Hills, 1, bld. 3, 119991 Moscow, Russia; annaplutalova@gmail.com; 3A.V. Topchiev Institute of Petrochemical Synthesis of Russian Academy of Sciences, Leninsky Av., 29, 119991 Moscow, Russia

**Keywords:** acrylonitrile copolymers, cyclization reaction, thermal oxidative stabilization, melt processable polyacrylonitrile

## Abstract

Binary and ternary copolymers of acrylonitrile (AN), *tert*-butyl acrylate (TBA), and *n*-butyl acrylate (BA) are synthesized through conventional radical polymerization in DMSO in the presence of 2-mercaptoethanol. The thermal behavior of binary and ternary copolymers is studied under argon atmosphere and in air. It is demonstrated that the copolymers of AN contain 1–10 mol.% of TBA split isobutylene upon heating above 160 °C, resulting in the formation of the units of acrylic acid in the chain. The carboxylic groups formed in situ are responsible for the ionic mechanism of cyclization, which starts at lower temperatures compared with pure polyacrylonitrile (PAN) or AN copolymer with BA. The activation energy of cyclization through ionic and radical mechanisms depends on copolymer composition. For the ionic mechanism, the activation energy lies in the range ca. 100–130 kJ/mole, while for the radical mechanism, it lies in the range ca. 150–190 kJ/mole. The increase in the TBA molar part in the copolymer is followed by faster consumption of nitrile groups and the evolution of a ladder structure in both binary and ternary copolymers. Thus, the incorporation of a certain amount of TBA in PAN or its copolymer with BA allows tuning the temperature range of cyclization. This feature seems attractive for applications in the production of melt-spun PAN by choosing the appropriate copolymer composition and heating mode.

## 1. Introduction

Copolymers of acrylonitrile (AN) with various vinyl monomers have wide applications in different fields as standard plastics and textile fibers. They are also used as precursors for carbon fibers [1,2,3,4,5]. In the latter case, fibers produced from copolymers based on polyacrylonitrile (PAN) undergo thermal oxidative stabilization (TOS), followed by carbonization in an inert atmosphere, resulting in the production of carbon fibers [6,7,8,9,10]. The mechanical properties of PAN-based precursors and final carbon fibers depend on numerous factors, including the molecular structure of the polymer determined by the polymerization conditions, the fiber morphology determined by spinning conditions, and the defects in the graphite-like structure determined by TOS and carbonization conditions.

Among the monomers used in the production of PAN-based copolymers, alkyl acrylates, and in particular, methyl acrylate, are the most commonly used in the synthesis of binary and ternary AN copolymers [11,12,13,14,15,16,17,18,19,20]. Binary AN–alkyl acrylate copolymers can be prepared via radical polymerization in bulk [21], solution [22], emulsion [23], or precipitation polymerization in water [24]. In solution radical polymerization, AN is more reactive than alkyl acrylate, and the reactivity of alkyl acrylate in copolymerization with AN decreases as the length of the alkyl substituent increases [25]. Alkyl acrylates do not exhibit an accelerating or inhibiting effect on the cyclization rate, while they participate in the chain transfer reaction resulting in the break of the development of the ladder structure and the initiating of cyclization in another part of the macromolecule (Figure 1) [25].

Since alkyl acrylates are inert in the cyclization reaction (alkyl = methyl, ethyl, *n*-butyl, 2-ethylhexyl, and lauryl), they do not affect the activation energy of the cyclization reaction at least at 10 mol.% of alkyl acrylate in the copolymer [25]. However, as the length of the alkyl substituent increases, the formation of the ladder structure occurs more slowly. In contrast, under conditions of thermal oxidative stabilization, the stabilization index increases as the number-average sequence length of the acrylonitrile units in the copolymers increases. Additionally, the incorporation of alkyl acrylate in PAN results in a decrease in glass transition temperature, while this effect becomes more pronounced as the length of the alkyl substituent in alkyl acrylate increases [25].

The content of acrylate groups in PAN-based copolymers used to produce carbon fiber precursors depends on the method of fiber molding. In wet spinning, the copolymer may contain any amount of alkyl acrylate comonomer. However, amounts of 1–5 mol.% is preferable for obtaining a low-defect fiber structure along with a high-molecular-weight polymer. In melt spinning, the melting point of pure PAN exceeds 300 °C, while the cyclization reaction (Figure 2) takes place at a lower temperature (approximately 260–270 °C) before the melting process occurs [26,27,28,29]. Thus, direct melt spinning of high-molecular-weight PAN becomes impossible because of the cyclization reaction occurring below 300 °C and the polymer melting above this temperature.

This dilemma can be solved by reducing the melting point below the cyclization temperature by decreasing the interaction between the nitrile groups in the AN units and decreasing the molecular weight of the polymer. This can be achieved by chemical and/or physical methods. The chemical approach involves using a copolymer of AN with a flexible comonomer taken in a sufficient amount to disrupt the chain regularity and to reduce the melting point to an acceptable level [30]. The physical approach involves plasticizing the polymer with an external plasticizer [31].

The properties of carbon fibers produced from wet-spun binary and ternary copolymers of AN and alkyl acrylates have been discussed in many publications and reviews [32,33,34,35]. Less attention has been paid to melt-spun copolymers. Binary copolymers of AN and methyl acrylate (MA) synthesized by precipitation polymerization in a DMF/water mixture with a number-average molecular weight (M_n_) of ~20 kg·mole^−1^ can melt at an MA content above 10 mol.% [36]. The flow activation energy decreases, and the melt lifetime increases with an increase in the MA content in the copolymer. Copolymers of AN and MA obtained from similar monomer feeds by various polymerization methods (solution polymerization in DMF, precipitation, and suspension polymerizations in water) exhibit different behaviors during melting [37]. An analysis of data presented in [36,37,38,39] suggests that the main factor influencing the viscosity of a melt is the MWD (the presence of the high molecular weight fraction) of the copolymer in addition to its composition. The melting of AN and MA copolymers with M_n_ ~ 20 kg·mole^−1^ and MA content of 10–15 mol.% occurs at temperatures between 200 and 210 °C, while TOS lasts for about one day at 220 °C [40]. Copolymers with an M_n_ of 110–130 kg·mole^−1^, dispersity *Ð* = *M_w_*/*M_n_* of 2.2–2.8, and similar composition melt at 250 °C, but the stability of the melt does not exceed 4 min [39]. The binary copolymers of AN/*n*-butyl acrylate (BA) (5–20 mol.%) exhibit more pronounced effect of shifting the onset of cyclization to higher temperatures and reducing the intensity of heat flow than for AN/methyl acrylate copolymers [25]. The decrease in the conversion of the nitrile group and stabilization index at 225 and 250 °C in both argon and air atmospheres was observed, at least for copolymers containing 10 mol.% of *n*-butyl acrylate. These copolymers can also be considered prominent candidates for melt-spun technology as AN/MA copolymers. However, if the polymer is melted at a temperature before the start of cyclization, then the fiber will lose its shape completely or partially, turning into a melt during TOS. To solve this problem, several approaches have been developed [41,42,43]. In particular, to shorten the TOS duration, a small amount of a third monomer, up to 1–5 mol.%, able to accelerate cyclization can be introduced into the copolymer. These terpolymers contain ~13 mol.% MA and up to 3 mol.% itaconic acid or 4 mol.% of acrylamide, acrylic, or methacrylic acids provide stable melts at 200 °C. A similar idea was discussed in [44] for AN/BA/fumaronitrile, AN/2-ethylhexyl acrylate/fumaronitrile terpolymers, and in [39,45,46] for AN/MA/dimethyl itaconate terpolymer.

We assume that *tert*-butyl acrylate (TBA) may have a similar effect on TOS as acidic comonomers because of the ability of poly(*tert*-butyl acrylate) to split isobutylene and convert into polyacrylic acid on thermal treatment [47]. In the present study, we, for the first time, have performed the synthesis of binary AN/TBA and ternary AN/TBA/BA copolymers (Figure 1) in a DMSO solution in the presence of a chain transfer agent to control the molecular weights of the copolymers. We have shown that the carboxylic groups are formed in situ on heating the polymers resulting in the acceleration of the cyclization reaction.

## 2. Materials and Methods

### 2.1. Materials and Polymer Synthesis

Monomers, namely acrylonitrile (AN, 99%, Aldrich, St. Louis, MO, USA), *n*-butyl acrylate (BA, 99%, Aldrich), and *tert*-butyl acrylate (TBA, 99%, Aldrich) were distilled under atmospheric or reduced pressure before use. Potassium persulfate (PSK, 98%, Aldrich) and 2-mercaptoethanol (ME, 98%, Aldrich) used as an initiator and a transfer agent, respectively, were used as received. DMSO and DMF (99%, Fluka) were distilled under reduced pressure before use.

Radical copolymerization was conducted at 55 °C in an argon atmosphere in a 100 mL three-necked flask equipped with an overhead stirring device with an anchor-type mixer. A solution containing the calculated amount of initiator and transfer agent in DMSO was placed in the flask, and the mixture of AN and alkyl acrylate was loaded. The reaction mixture was bubbled with argon for 30 min, and the flask was placed into a preheated 55 °C bath. The formulations of the reaction mixtures are given in Table 1. In all syntheses, the total concentration of monomers in the solution was 20 wt%. After polymerization the reaction mixtures were cooled rapidly to room temperature, diluted with DMSO, if necessary, and then precipitated in an excess of water. The precipitate was filtrated, washed with water and methanol, and then dried in a vacuum to a constant weight. The monomer conversion was determined by gravimetry.

### 2.2. Instrumentation

The average molecular weights and dispersity (*Ð* = *M_w_*/*M_n_*) were determined by size exclusion chromatography (SEC). The SEC measurements were performed in DMF containing 0.1 wt% of LiBr at 50 °C with a flow rate of 1.0 mL/min using a chromatograph GPC-120 “PolymerLabs” (*Hichrom* Limited, UK) equipped with refractive index and with two columns PLgel 5 µm MIXED B for MW range 5 × 10^2^–1 × 10^7^. The SEC system was calibrated using narrow dispersed linear poly(methyl methacrylate) standards with MW ranging from 800 to 2 × 10^6^ g mol^−1^. A second-order polynomial was used to fit the log_10_*MW* versus retention time dependence.

Thermogravimetric analysis (TGA) was carried out with a synchronous thermal analyzer STA 449 F3 Jupiter (Netzsch, Selb, Germany). The mass loss of a 5–10 mg copolymer sample placed in a corundum crucible was analyzed in the linear heating mode (10 °C/min from 25 to 600 °C) under an argon flow of 50 mL/min or in air.

Fourier transform infrared (FTIR) spectroscopy in ATR mode (diamond crystal) was recorded using a Spectrum Two FT-IR Spectrometer (PerkinElmer, Waltham, MA, USA) in the range of 4000–600 cm^−1^. Samples were prepared by dissolving 4 wt% of copolymer in DMSO and cast onto a glass substrate. The samples were then placed in the vacuum oven at 80 °C overnight to remove DMSO. The resulting films were separated from the surface and analyzed. For quantitative analysis of the composition of the films of the synthesized copolymers, the calibration curves were used [25]. Calibration curves were obtained from the mixtures of the monomers of a known molar ratio. The ratio of the intensities, A of the characteristics bands assigned to ν_CN_ = 2229 cm^−1^ (AN) and ν_C=O_ = 1721 (BA and TBA) was used. In all the cases, the dependences of A_CO_/A_CN_ on the molar ratio [butyl acrylate]/[AN] were linear [25].

Differential scanning calorimetry was performed on a Netzsch DSC 204 (Netzsch, Germany) in the atmosphere of dry gas (air, argon) at a flow rate of 100 mL/min in the range of 30–500 °C and a heating rate from 5 to 20 °C/min in air and 10 °C/min in argon.

The activation energy of the reactions was determined by Kissinger method [48]:(1)$$-\frac{E_{a}}{R}=\frac{d\left[ln\left(\frac{\varphi }{T_{p}^{2}}\right)\right]}{d\left(\frac{1}{T_{p}}\right)}$$
where *R*—universal gas constant, *T_p_*—peak temperature (K), *ϕ*—heating rate (°C/min).

FTIR spectra in ATR mode (diamond crystal) of the films subjected to thermal treatment in air (at 180, 200, 225, and 250 °C) and in argon (at 225 and 250 °C) were recorded using Spectrum Two FT-IR Spectrometer (PerkinElmer) in the range of 4000–400 cm^−1^.

The proportion of the unreacted nitrile groups *φ*_CN_ [49] and stabilization index *E_s_* were determined according to the equations [50]:(2)$$\varphi_{CN}=\frac{A_{2240\;{\mathrm{cm}}^{-1}}}{A_{2240\;{\mathrm{cm}}^{-1}}+fA_{1590\;{\mathrm{cm}}^{-1}}}$$
(3)$$E_{s}=\frac{A_{1590\;{\mathrm{cm}}^{-1}}}{A_{2240\;{\mathrm{cm}}^{-1}}}$$
where $A_{2240\;{\mathrm{cm}}^{-1}}$—intensity of the band assigned to –C≡N group, $A_{1590\;{\mathrm{cm}}^{-1}}$—intensity of the band assigned to −C=N− group, f—the ratio of the extinction equal to 0.29 [49].

## 3. Results and Discussion

### 3.1. Binary Copolymers of Acrylonitrile and Tert-Butyl Acrylate

#### 3.1.1. Polymer Synthesis and Characterization

Copolymerization of AN and TBA initiated by a radical initiator, PSK, was carried out in DMSO at an overall monomer concentration of 20 wt %, with a range of TBA molar part in the monomer feed from 1 to 10 mol.%, in the presence of varying amounts of a chain transfer agent, ME. The copolymerization proceeded to high monomer conversions, and its rate was insensitive to the TBA content at the chosen monomer feeds (Figure 2a). The copolymer composition was analyzed using FTIR spectroscopy. The mole fraction of TBA in the copolymer increased with the mole fraction of TBA in the initial monomer mixture, while the average composition of the copolymer remained constant throughout the reaction (Figure 2b). Previously, we have demonstrated that TBA and BA have similar reactivity in radical copolymerization [51]. In addition, we have shown that RAFT copolymerization of AN—BA and AN—TBA in bulk has similar features [21]. Thus, it can be assumed that TBA and BA have similar reactivity in radical copolymerization with AN. Based on the reactivity ratios of AN and BA in DMSO previously determined [25] (r_BA_ = 0.63 and r_AN_ = 0.99) we estimated the instantaneous triad composition (*F*_AN-AN-AN_, *F*_AN-AN-TBA_, and *F*_AN-TBA-AN_) and number-average sequence lengths of AN (〈*N*_AN_〉*_n_*) and TBA (〈*N*_TBA_〉*_n_*) (Table 1) using the Equations (4)–(8) [52].
(4)$$F_{\mathrm{AN}-\mathrm{AN}-\mathrm{AN}}={\left(1-\frac{1}{1+\frac{r_{\mathrm{AN}}f_{\mathrm{AN}}}{f_{\mathrm{TBA}}}}\right)}^{2}$$
(5)$$F_{\mathrm{AN}-\mathrm{AN}-\mathrm{TBA}}=\frac{2}{1+\frac{r_{\mathrm{AN}}f_{\mathrm{AN}}}{f_{\mathrm{TBA}}}}\left(1-\frac{1}{1+\frac{r_{\mathrm{AN}}f_{\mathrm{AN}}}{f_{\mathrm{TBA}}}}\right)$$
(6)$$F_{\mathrm{AN}-\mathrm{TBA}-\mathrm{AN}}={\left(\frac{1}{1+\frac{r_{\mathrm{AN}}f_{\mathrm{AN}}}{f_{\mathrm{TBA}}}}\right)}^{2}$$
(7)$${\langle N_{AN}\rangle }_{n}=\frac{1}{1-p_{AA}}\;\mathrm{and}\;{\langle N_{TBA}\rangle }_{n}=\frac{1}{1-p_{BB}}$$
where p_AA_ is the probability that chain terminus AN adds to AN monomer and p_BB_ is the probability that chain terminus TBA adds to TBA monomer:(8)$$p_{AA}=\frac{r_{AN}f_{AN}}{f_{AN}\left(r_{AN}-1\right)+1}\;\mathrm{and}\;p_{BB}=\frac{r_{TBA}f_{TBA}}{f_{TBA}\left(r_{TBA}-1\right)+1}$$

The composition of triads and the number-average sequence length of AN and TBA units are summarized in Table 2. The results show that within the range of TBA molar ratios in the monomer feed, between 1.0 and 10.0 mol.%, F_AN-AN-AN_ triads dominate the composition of the macromolecules. The number-average sequence length of TBA units is equal to 1, indicating that the sequences of AN units are often interrupted by single TBA units. This microstructure of the chains should affect the thermal properties of the synthesized copolymers.

#### 3.1.2. In Situ Formation of Carboxylic Groups during Pyrolysis of AN–TBA Copolymer

The thermal behavior of AN–TBA copolymers differs from that of AN copolymers containing alkyl acrylates, such as methyl-, ethyl, and *n*-butyl [25]. Figure 3 demonstrates the weight loss of pure PAN (curve 1) and AN–TBA copolymers of various compositions (curves 2–5). The incorporation of TBA causes a shift in the temperature at which weight loss begins to occur to a lower temperature region (below 200 °C). Additionally, a new step in weight loss appears on the curves, becoming more pronounced with increasing TBA content in the copolymer. This weight loss may be attributed to the release of isobutylene from the *tert*-butyl acrylate unit. This assumption is supported by data in Table 3, which shows that the weight loss of the copolymer results from the release of isobutylene. For comparison, TGA data for the degradation of pure polyacrylic acid is also provided (curve 6). As can be seen, polyacrylic acid is less stable under heating than AN–TBA copolymers due to the presence of carboxylic groups that can split water and CO_2_.

The mechanism of poly(*tert*-butyl acrylate) pyrolysis has been discussed in numerous papers [47,53,54]. The non-radical nature of the process was confirmed by pyrolysis of a sample of polymer in the presence of a radical spin-trap. The pyrolysis mechanism involves the formation of a 6-membered ring intermediated, which results in the release of isobutylene and the formation of polyacrylic acid (Figure 3). However, the presence of nearby carboxylic groups may cause an autocatalytic pyrolysis reaction and even the formation of anhydride groups (Figure 4).

In the synthesized copolymers of AN and TBA, the TBA units are mostly surrounded by AN units (Table 2). Therefore, autocatalytic pyrolysis and the formation of intramolecular anhydride groups seem unlikely. To investigate the changes in the chemical structure of the copolymers, we conducted two experiments. In the first experiment, 5 wt % DMSO solutions of copolymers were synthesized from a monomer feed containing 5.0, 7.5, and 10.0 mol.% of TBA were heated at different temperatures (140, 160, and 180 °C) for the required period. After that, DMSO was evaporated at 100 °C under low pressure to obtain films. These films were then analyzed using FTIR spectroscopy (Figure 4). Upon heating, all the solutions changed the color from colorless to pale yellow at 140 °C and brown at 180 °C. However, the copolymers remained soluble in DMSO after heating. Notably, changes in the FTIR spectra were observed in the ranges of ca. 3500, 1700, and 1200 cm^−1^. These changes were more pronounced at higher temperatures and for longer heating times. Upon heating, the absorption band at 1730 cm^−1^, corresponding to the stretching vibrations of the carbonyl group (ν_C=O_) in TBA, shifts to 1720 cm^−1^ and then splits into three bands with maxima at 1716, 1694, and 1672 cm^−1^. At the same time, a low-intensity band appears at 1540 cm^−1^. These new bands are attributed to monomeric, dimeric, and anionic forms of the carboxylic group. The broad absorption bands at ca. 3500 and 2600 cm^−1^ are due to free and bound OH groups, respectively. Therefore, we have confirmed the formation of carboxylic groups in the copolymers upon heating. The appearance of the color is the consequence of the conjugation in the polymer. However, its solubility suggests a low degree of conjugation. Therefore, it can be assumed that carboxylic groups can initiate the cyclization of nitrile groups, which is consistent with the emergence of new absorption bands in the range of 1500–1000 cm^−1^. It is worth noting that the ratio of absorption band intensities referring to CN groups vibrations and CH_2_ groups of the main chain (A_2240_/A_1450_) remains nearly constant, confirming a low degree of conjugation.

In the second experiment, films were cast from a copolymer solution in DMSO, and the dried films were subjected to thermal treatment at a constant temperature (Figure 5, Appendix A). The samples changed color more rapidly than upon heating in DMSO solution, and lost solubility during this process. We believe this is due to an intermolecular reaction between the carboxylic groups of different macromolecules, which leads to the formation of an anhydride that links macromolecules together. Figure 5 shows the FTIR spectra of the films made from the copolymers containing different amounts of TBA, heated at 180 °C for a certain period. Other spectra from thermal treatments at 140 and 160 °C are provided in the ESI (Appendix A). After heating, the absorption band corresponding to the stretching vibrations of the carbonyl group (ν_C=O_) in TBA splits into three bands with maxima at 1760, 1725, and 1697 cm^−1^. New absorption bands also appear at 1803 and 1044 cm^−1^, which are attributed to the formation of an anhydride [53]. Changes in the FTIR spectra in the region of ca. 3500, 2600, and 1500–1000 cm^−1^ are less pronounced compared with the reaction in DMSO solution. Therefore, intermolecular formation of anhydride becomes preferable during reaction in bulk, resulting in the retardation of cyclization compared with the reaction in solution.

These experiments have shown that when copolymers of AN and TBA are heated, carboxylic groups form in situ due to the release of isobutylene. These carboxylic groups can either initiate the cyclization of nitrile groups or convert into anhydrides and link macromolecules. As a result, these transformations can influence the thermal behavior of AN–TBA copolymers during the process of cyclization and thermal oxidative stabilization.

#### 3.1.3. Thermal Behavior under Argon Atmosphere

DSC was applied in the dynamic conditions of heating to investigate the thermal properties of AN–TBA copolymers. In an argon atmosphere, the cyclization of PAN results in intense heat release in the narrow temperature range (Figure 6a, curve 1). The incorporation of TBA units results in a shift in the temperature, at which exo-effect begins, to the lower-temperature region and a decrease in the intensity of heat flow (Figure 6a, curves 2–6, Table 4). An increase in the TBA content in the copolymers leads to a broadening of the temperature range of the exo-effect. The analysis of DSC curves of the copolymers reveals three peaks at ca. 220 (peak I), 260–270 (peak II), and 280–290 °C (peak III). Additionally, an endo-effect at ca. 200 °C is observed, which is more pronounced for the copolymer synthesized from the monomer mixture containing 10.0 mol.% of TBA. This endo-effect is attributed to the release of isobutylene from TBA units. Peak I may be assigned to the formation of an anhydride due to the reaction of carboxylic groups formed in situ after the release of isobutylene. Its locus on the temperature scale is independent of the TBA content in the copolymer, while the intensity of heat flow increases with an increase in TBA content in the copolymer (Figure 6a, Table 3). Peak II can be assigned to the ionic cyclization of nitrile groups by the carboxylic groups (Figure 5). Peak III is due to cyclization initiated by a radical mechanism. This assumption is based on the results of our previous studies of the cyclization behavior of AN copolymers with acrylic acid [55,56]. A similar trend is observed for copolymers with lower MWs (Figure 6b). The temperature range corresponding to the reactions proceeding with exo-effect increases with the growth of TBA content in the copolymer, while TBA content has no visible impact on the total enthalpy of these reactions (Table 4).

An increase in the TBA content in the copolymer results in an increase in the intensity of heat flow corresponding to peak II and a decrease in the intensity of heat flow corresponding to peak III, which accords with a rise in acrylic acid content and our previous studies [55,56]. The activation energy of cyclization due to ionic (peak II) and radical mechanism (peak III) was estimated from DSC thermograms of copolymers recorded in argon at various ramp rates (Figure 7 and Appendix A). Processing of these data by Equation (1) results in linear dependences of $ln\left(\varphi /T_{p}^{2}\right)$ on 1/*T_p_* (Appendix A). The values of activation energy of the cyclization reaction calculated from these data are summarized in Table 4. These results accord well with our previous findings for the copolymers of AN and acrylic acid [55,56]. The activation energy of ionic cyclization is close to the values typical for copolymers of AN and acrylic acid: 100–140 kJ/mole (ionic cyclization) and 160–230 kJ/mole (radical cyclization) [56].

The cyclization reaction is accompanied by changes in the chemical structure of macromolecules, which can be followed by FTIR spectroscopy. Figure 8 presents the typical FTIR spectra of copolymer films subjected to heating at 200 (a) and 225 °C (b). FTIR spectra of other copolymers with various TBA contents are shown in Appendix A.

After heating the films, new absorption bands appeared in the spectra at 3350 (ν_NH_), 1581–1650 (ν_-C=N-_, ν_N-H_, ν_C=O_), 1485 (ν_-C=N-_) and 1380 (δ_-C-H, C-H_), 1246 and 1150 cm^−1^ (ν_C-C, C-N, C-O_). These bands can be attributed to new bonds formed in the conjugation system. Over time, the intensity of these bands increased, while the intensity of the absorption band for the stretching vibrations of the nitrile group (ν_C≡N_) decreased. All these changes indicate the formation of a well-developed polyconjugated system due to the cyclization of nitrile groups. The intensity of the absorption bands for the bending vibrations of CH_2_ groups in the main chain of δ_CHH_ (at 1450 cm^−1^) and the stretching vibrations for CH (2940 cm^−1^ (ν_sC−H_) and 2870 cm^−1^ (ν_asC−H_)) did not change, indicating the absence of dehydrogenation reaction. Notably, when heated, the intensity of the absorption band corresponding to the ester group (ν_C=O_) decreases, and at the same time, a new band appears at ~1693 cm^−1^. This can be attributed to the vibrations of the carbonyl group of the naphthyridine and acridone rings formed by the interaction of the carboxyl and nitrile groups in the copolymer.

The analysis of FTIR spectra of polymers heated at 200 and 225 °C allows for quantitative estimation of the cyclization reaction. The proportion of the unreacted nitrile groups *φ*_CN_ was determined according to Equation (2). The dependence of *φ*_CN_ on the time of heat treatment at different temperatures is plotted in Figure 9. At 200 °C, the rate of conversion of nitrile groups increased with an increase in the TBA content in the copolymer, and it was higher compared with pure PAN (Figure 9a). This could be due to the initiation of the cyclization reaction by carboxylic groups that are formed in situ. This cyclization occurs via an ionic mechanism at lower temperatures than initiation via a radical mechanism, which is a common cyclization mechanism for PAN. The rise in the temperature should result in an increase in the rate of the formation of an anhydride through intermolecular reactions of carboxylic groups. Additionally, the contribution of radical cyclization increased with rising temperature, and the rate of nitrile group conversion slightly depended on the TBA content (Figure 9b). These findings agree with DSC data. The stabilization index, E_s_, is another quantitative parameter used to estimate the efficiency of the cyclization reaction. It can be calculated using Equation (3). From Figure 9c,d, we can see that at a lower temperature of 200 °C, the E_s_ for AN–TBA copolymers increase more rapidly compared with pure PAN. However, this trend decreases at a higher temperature of 225 °C. In general, the E_s_ values are similar to those of AN–alkyl acrylate copolymers rather than AN–acrylic acid copolymers [25,55].

#### 3.1.4. Thermal Oxidative Stabilization

Thermal oxidative stabilization is used for the production of carbon fibers instead of thermal treatment in an inert atmosphere. However, this process is more complicated, and along with the cyclization, it includes dehydration (+O_2_; –H_2_O) and oxidation reactions. Figure 10 shows DSC curves of AN–TBA copolymers recorded in air under heating with a rate of 10 °C/min. The appearance of new reactions leads to more intense heat flow compared with experiments in an argon atmosphere. The incorporation of TBA into the PAN chain, even at low content, results in a noticeable decrease in the intensity of heat flow and shift thermograms in the lower-temperature region. The higher the TBA content, the broader the temperature range at which exo-effect is observed and the lower the intensity of heat flow (Table 5). Comparison of Figure 6 and Figure 10 leads to the assumption that the lower-temperature peaks observed during TOS can be attributed to the cyclization reactions, while high-temperature peaks are attributed to the dehydration and oxidation processes.

To prove this assumption, we have analyzed the changes in the chemical structure of copolymers under heating at 200 (a) and 225 °C (b) by FTIR spectroscopy (Figure 11 and Appendix A). A comparison of Figure 8 and Figure 11 shows that FTIR spectra of the copolymers containing 10.0 mol.% TBA and heated at 200 and 225 °C under argon and air atmospheres are similar. Therefore, the cyclization reaction prevails over the dehydration and oxidation processes in the lower-temperature region.

Evidently, the conversion of nitrile groups and the stabilization index have a similar trend (Figure 12) as in the case of heat treatment in argon (Figure 9). However, upon heating in air at 200 °C, the polyconjugated system is developed faster than in argon, while at 225 °C the rates of both processes are similar.

Therefore, we have demonstrated that AN copolymers containing TBA are capable of in situ formation of carboxylic groups due to the splitting of isobutylene under heating above 160–180 °C. These carboxylic groups cause initiation of cyclization of nitrile groups already at 200 °C and developing of polyconjugated structure. However, the rate of this process is lower compared with AN–acrylic acid copolymer [56].

### 3.2. Ternary Copolymers of Acrylonitrile, Tert-Butyl Acrylate, and n-Butyl Acrylate

The thermal behavior of AN copolymers with BA differs from that of AN copolymers with TBA, which can be illustrated in Figure 13a. The increase in BA content in the copolymer results in a shift in the thermograms to the high-temperature region and a decrease in the intensity of heat flow. These effects come from the ability of alkyl acrylates to participate in the chain transfer reaction, which results in the break of the development of the ladder structure and the formation of another polymeric radical able to initiate cyclization (Figure 1) [25].

When both TBA and BA units are incorporated into the PAN macromolecule, the polymer reveals hybrid thermal behavior (Figure 13b, Table 6). Exo-effect is observed at higher temperatures if compared with the AN–TBA copolymers (Table 4), while the temperature range at which exo-effect is observed is broader. The higher the TBA content in the copolymer, the more intense the exo-effect in the low-temperature region. It can be assumed that both acrylate monomers affect the thermal behavior of the AN copolymer.

A similar conclusion can be obtained from TGA data (Figure 14). The incorporation of BA units into the PAN macromolecule shifts the beginning of the weight loss to the high-temperature region, while the incorporation of TBA units shifts the beginning of the weight loss to the lower-temperature region (Table 7).

It can be assumed that the rate of the development of the polyconjugated system in terpolymers under thermal treatment will be intermediated between corresponding binary copolymers of AN–BA and AN–TBA. Figure 15 shows FTIR spectra of AN–BA–TBA terpolymers synthesized from the mixture containing 15 mol.% BA and 5 mol.% TBA and heated at 200 and 225 °C under an argon atmosphere. FTIR spectra of other terpolymers are given in the ESI file (Appendix A). At 200 °C, the changes in the FTIR spectra are negligible, indicating the low rate of cyclization reaction. This accords well with our previous findings for AN–BA copolymers. The rate of cyclization increases when the films are heated at 225 °C. Nevertheless, the presence of TBA units accelerates the cyclization reaction when compared with AN–BA copolymers (Figure 16).

A similar trend is observed during thermal oxidative stabilization (Figure 17 and Figure 18).

Therefore, the incorporation of both acrylate units, BA and TBA, allows for the broadening of the temperature range corresponding to the formation of the ladder structure. Their ratio in the AN terpolymer allows for control of the cyclization reaction rate. Increasing the TBA/BA ratio results in an increase in the cyclization reaction rate and vice versa.

## 4. Conclusions

The results of our studies have revealed the effect of *tert*-butyl acrylate on the thermal behavior of the binary and ternary copolymers of acrylonitrile with relatively low MW, which is required for the production of melt processable PAN. The incorporation of TBA units in PAN results in noticeable changes in the cyclization behavior of PAN. These changes come from the ability of TBA units to split isobutylene, which results in the formation of carboxylic groups in AN copolymer. As formed carboxylic groups are capable of initiating the cyclization reaction via an ionic mechanism at lower temperatures in comparison with pure PAN that undergoes cyclization via a radical mechanism. Therefore, in situ formation of carboxylic groups results in the broadening of the temperature range, at which the formation of ladder structure occurs. The additional incorporation of BA makes the temperature range of cyclization even wider.

We have also shown that the synthesized copolymers containing a low amount of TBA lose their solubility after continuous heating (24 h) at 140 °C because of the intermolecular reaction of the formation of anhydride groups. Therefore, it can be assumed that the melt processing of an AN copolymer with TBA units should be carried out below 140 °C, followed by TOS, which should occur at temperatures above this value. Preliminary results indicate that direct melting of AN copolymers containing TBA with a M_n_ ca. 19–27 kg·mole^−1^ is unlikely, suggesting the need for the use of a plasticizer.

## Data Availability

The data presented in this study are available on request from the corresponding author. The data are not publicly available due to privacy.

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
