# Peer review of "In Situ Formation of Acidic Comonomer during Thermal Treatment of Copolymers of Acrylonitrile and Its Influence on the Cyclization Reaction"

_polymers, 2024, doi:10.3390/polym16192833_

Round 1
Reviewer 1 Report
Comments and Suggestions for Authors
attached file

attached file
Author Response
We would like to express our sincere gratitude to the reviewer for the helpful comments and suggestions. We have carefully revisited the article in accordance with all the comments. All the noticeable corrections are marked in red in the text.
Comment 1: The authors used the bulk of reference lines (44-45) used them properly.
Reply: All the references 11 – 20 are devoted to the binary and ternary copolymers of acrylonitrile and alkyl acrylates, mostly methyl acrylate.
Comment 2: how many samples for each category were used to measure FTIR spectrums, DSC, TGA? It would be great to provide the film pictures Upon heating all the solutions changed color from colorless to pale yellow at different temperatures (Lines 251-254).
Reply: The prepared polymer films were analyzed by FTIR spectroscopy, DSC and TGA. 3 – 5 FTIR spectra were recorded for each sample from different places of the film and the average data were used in analysis of the conversion of nitrile group. DSC was reproduced several times: first heating to c.a. 130 – 150 oC, then cooling to room temperature and after that heating again. The results of the second heating are presented in the paper. TGA was reproduced twice. The films also have changed their color as it was indicated in the text. We have added the phrase that color was changed more rapidly than upon heating in DMSO solution.
Comment 3: In contrast, under thermal oxidative stabilization (TSO) conditions, the stabilization index (Line 60) Please check the abbreviation throughout the manuscript.
Reply: We have checked the text, two terms, namely thermal oxidative stabilization and the stabilization index are used throughout the paper as well as abbreviation TOS.
Comment 4: Line 88 MA means what? Define first then use the abbreviation these
Reply: MA is a methyl acrylate. We have added a transcript of the abbreviation
Comment 5: Co-polymers with Mn of 110–130 kgmol-1, Đ of 2.2–2.8, and similar composition melt at 250°C, but the stability of the melt does not exceed 4 min. rewrite this sentence and explain the unit first.
Reply: Mn is a number-average molecular weight and Đ is a dispersity equal to Mw/Mn. The dimension of Mn is kg/mole (or g/mole) or kDa (Da). We have corrected and replaced mol by mole.
Comment 6: A second-order polynomial was used to fit the log10 M check this in the whole article.
Reply: log10M was replaced by log10MW
Reviewer 2 Report
Comments and Suggestions for Authors
The submitted manuscript describes preparation of (1) copolymers of acrylonitrile and t-butyl acrylate, and (2) terpolymers of acrylonitrile, t-butyl acrylate, and n-butyl acrylate by radical polymerization and their behaviour under heating in argon atmosphere and in air. I recommend its publication after following questions/comments being responded.
1. How the formation of isobutylene (Table 3) was identified?
2. The equations (2) and (3) for CN group determination need to be clarified: (a) the reference given there do not concern FT IR, (b) it is necessary to show that these equations can be applied also for ATR technique.
3. As concerns φCN determination using foregoing equations: the peak of CN group is base-line separated, however, the peak of -C=N- group is overlapped with other peaks. How its value A was evaluated?
4. Fig. 12. – description of individual curve (1-5) is missing.
5. The statements (line 42, p. 15) “the rate of this process is lower compared to AN – acrylic acid copolymer“ need probably a reference.
6. The first paragraph on page 13 seems to be not clear, please check the correspondence of the text with Fig. 9.
7. If I understood well the composition of terpolymers was not determined. It is characterised by the composition of starting polymerization mixture. Do you think that the terpolymer composition is the same as this polymerization mixture?
8. Formal errors – a) P.14, line 40 - 2225 °C should be probably 225 °C, b) Table 1, last row – the data are not for FTBA but FBA, probably.
Author Response
We would like to express our sincere gratitude to the reviewer for the helpful comments and suggestions. We have carefully revisited the article in accordance with all the comments. All the noticeable corrections are marked in red in the text.
Comment 1. How the formation of isobutylene (Table 3) was identified?
Reply. In this research, we have not identified directly isobutylene. However, by means of FTIR spectroscopy we have determined the formation of carboxylic groups after thermal treatment. The formation of carboxylic groups results from the splitting of isobutylene, which accords with numerous literature data. Corresponding references are given in the text of the paper (ref. 47, 53,54)
Comment 2.The equations (2) and (3) for CN group determination need to be clarified: (a) the reference given there do not concern FT IR, (b) it is necessary to show that these equations can be applied also for ATR technique.
Reply. We have found a mistake in the reference. Instead of ref.49, ref.39 was given. Now, it is corrected. In ref.49, FTIR spectroscopy has been applied to analyze the transformation of nitrile groups during cyclization reaction and the value of f was determined. In our previous studies, we have confirmed that ATR technique can also be applied to estimation of conversion of nitrile groups as we use the ratio of the intensities of the absorbance bands instead of absolute values of bands intensities. Besides, we have analysed polymer films prepared from the common solvent, DMF. In all the experiments, 3 – 5 spectra have been recorded and the average result is given; the measurement error has not exceeded 5 %.
Comment 3. As concerns φCN determination using foregoing equations: the peak of CN group is base-line separated, however, the peak of -C=N- group is overlapped with other peaks. How its value A was evaluated?
Reply. The separation of the peaks was performed in Origin program, after that A was determined.
Comment 4. Fig. 12. – description of individual curve (1-5) is missing.
Reply. We have added the description of the curves: (a, c) fTBA = 1 (1), 2.5 (2), 5 (3), and 10 mol. % (4); (b, d) fTBA = 0 (1), 1 (2), 2.5 (3), 5 (4), and 10 mol. % (5).
Comment 5. The statements (line 42, p. 15) “the rate of this process is lower compared to AN – acrylic acid copolymer“ need probably a reference.
Reply. The reference 56 was added.
Comment 6.The first paragraph on page 13 seems to be not clear, please check the correspondence of the text with Fig. 9.
Reply. The references to the Figs. 9a and 9b were mixed up. Now they are corrected. We also have added a few comments to cyclization mechanism and added the references 25 and 55 to illustrate the differences in Es for AN/alkyl acrylate and AN/acrylic acid copolymers.
Comment 7. If I understood well the composition of terpolymers was not determined. It is characterised by the composition of starting polymerization mixture. Do you think that the terpolymer composition is the same as this polymerization mixture?
Reply. We can determine only the overall concentration of acrylate monomers in the copolymer. As the reactivity of BA and TBA is similar, we suppose that they are incorporated in the copolymer with the similar rate and their molar content is proportional to their content in the monomer feed.
Comment 8. Formal errors – a) P.14, line 40 - 2225 °C should be probably 225 °C, b) Table 1, last row – the data are not for FTBA but FBA, probably.
Reply. We have corrected. Thank you. In Table 1, the last row – the overall concentration of acrylate monomers is given. This information is added to notes.